



# An overlooked freshwater source contributed to the extreme freshening event in the eastern subpolar North Atlantic after 2014

Bogi Hansen[1*], Karin Margretha Húsgarð Larsen[1], Hjálmar Hátún[1], Steingrímur Jónsson[2], Sólveig Rósa Ólafsdóttir[3], Andreas Macrander[3], William Johns[4], N. Penny Holliday[5], Steffen Malskær Olsen[6]

[1]Faroe Marine Research Institute, Tórshavn, Faroe Islands.
[2]University of Akureyri, Akureyri, Iceland
[3]Marine and Freshwater Research Institute, Hafnarfjörður, Iceland.
[4]Department of Ocean Sciences, Rosenstiel School of Marine and Atmospheric Science, University of Miami, Coral Gables, FL, USA.
[5]National Oceanography Centre, Southampton, UK.
[6]Research and Development, Danish Meteorological Institute, Copenhagen, Denmark

*Correspondence to*: Bogi Hansen (bogihan@hav.fo)

**Abstract.** Outflows of low-salinity waters from the Arctic to the upper layers of the subpolar North Atlantic (SPNA) are central in redistributing freshwater from river runoff, melting sea ice, and precipitation. They act to reduce shallow, as well as deep, convection; thereby affecting both biological production and the Atlantic Meridional Overturning Circulation. The two main sources of low-salinity water to the SPNA are the flows through the Canadian Arctic Archipelago and through the

Denmark Strait. A potential additional source of low-salinity water is the shelf/slope region south of Iceland, mainly fed by Icelandic runoff. Normally this water passes into the Nordic Seas, but in some periods, it may instead flow into the upper layers of the central parts of the Iceland Basin in the eastern SPNA. This low-salinity water has previously been overlooked as a freshwater supply to the SPNA. Using a range of observational data sets, we show that the conditions for a diversion of this water mass from the south Iceland shelf into the Iceland Basin were favourable during the 2014–2018 period. In those

years the Iceland Basin became extraordinarily fresh, characterized by surface salinity lower than previously seen in a 120-year long time series. The event is thought to have been mainly caused by unusual winter wind stress patterns that diverted freshwater from the western SPNA to the eastern basin and caused a zonal shift of the subpolar front. Here, we show that the low-salinity signal near the surface was locally reinforced in the central Iceland Basin by anomalous diversion of low-salinity water originating in the shallow shelf areas south of Iceland and that this can help explain why the surface salinity of

the Iceland Basin became so exceptionally low. The diversion was generated by anomalous wind conditions over the Iceland Basin and caused slightly enhanced freshening of the warm waters crossing the Greenland-Scotland Ridge from the SPNA into the Nordic Seas. The low-salinity Icelandic-source water also increased the near-surface stratification and reduced the depth of convection in the Iceland Basin during two consecutive winters with reduced nutrient renewal of near-surface waters as a consequence. Although especially pronounced after 2014, this extra freshwater input probably occurs more

generally, which may help explain why the central Iceland Basin may be an oligotrophic region, as has previously been suggested.



# 1 Introduction

The subpolar North Atlantic (SPNA) acts as a buffering region between the Arctic Mediterranean (Nordic Seas plus Arctic Ocean with shelves) and the rest of the world oceans. From this region, warm and relatively saline water crosses the Greenland-Scotland Ridge into the Arctic Mediterranean while the outflows from the Arctic Mediterranean enter the region both near the surface on both sides of Greenland and as deep "overflow" east of Greenland (Hansen and Østerhus, 2000).

The overflow is the main source of high-density water for the lower limb of the Atlantic Meridional Overturning Circulation (AMOC), but during its passage through the SPNA, the overflow water is strongly modified by entrainment of ambient water. Salinity variations in the SPNA will therefore affect the densities of both the warm-water flow towards the Arctic and the water entrained into the overflow, thereby affecting thermohaline ventilation both north of and south of the Greenland-Scotland Ridge.

Inflow of low-salinity water will also affect the density stratification near the surface. By reducing vertical mixing, increased stratification may affect instantaneous primary production and may also reduce the depth of winter convection. Since winter convection is an important mechanism for nutrient replenishment of the euphotic zone, this process may reduce the long-term potential for primary production and hence also production at higher trophic levels.

Most of the low-salinity water enters the SPNA in its western regions (e.g., Tesdal and Haine, 2020), but the flow from the SPNA to the Arctic Mediterranean is from the eastern SPNA and most of the entrainment into overflow also occurs in the eastern SPNA (e.g., Lozier et al., 2019). Salinity variations in the eastern SPNA and their causes are therefore of special importance in a climatic perspective and periods of (negative) "salinity anomalies" in this region have received much interest (e.g., Dickson et al., 1988; Belkin et al., 2000). This study focuses on the freshening event that culminated in the eastern SPNA after 2014 and has been reported by Holliday et al. (2020). We will refer to this as the "post-2014 freshening event".

The development of this event may be illustrated by the salinity variations of the warm-water flow from the SPNA to the Arctic Mediterranean. This flow is structured into three different current branches, here labelled: the Iceland branch (I-branch), the Faroe branch (F-branch), and the Shetland branch (S-branch) (Fig. 1a). Monitoring of the hydrographic properties of these flows started more than half a century ago for the S-branch and all of the branches have been regularly monitored since 1991 with salinity values freely available in the ICES IROC database (Fig. 1b).

Since they are based on only a few hydrographic cruises each year, the reported annual salinity values of these branches have considerable uncertainties, but large changes are generally found to be robust (e.g., Larsen et al., 2012) and the consistency between the different branches (Fig. 1b) supports this. Thus, there can be little doubt that this flow started to freshen around 2010 with an especially rapid drop in salinity from 2015 to 2017. This salinity drop is seen in all the





branches, but increases as we move from the S-branch over the F-branch to the I-branch, for which we here use the Faxaflói

9 standard station (Casanova-Masjoan et al., 2020).

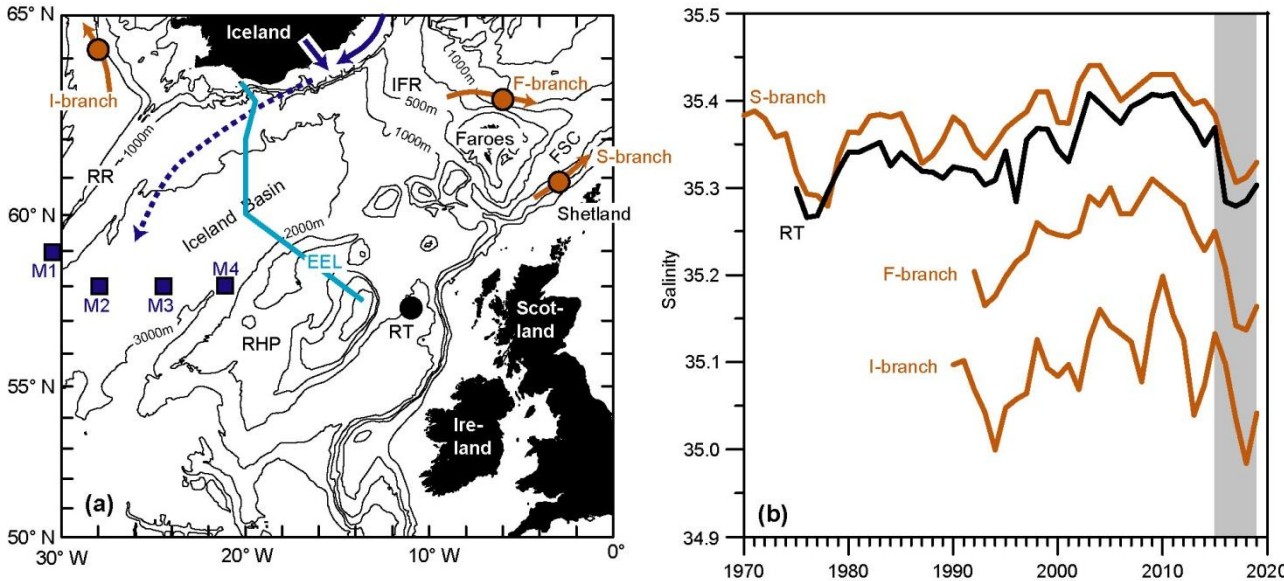

**Figure 1. (a)** Topographic features and observational sites in the eastern SPNA. Brown arrows indicate the three branches of warm-water flow to the Arctic Mediterranean with brown circles indicating locations where salinity is monitored for each of them. Black circle indicates the reference site in the Rockall Trough (RT). Continuous blue arrows indicate the additional freshwater input to the Iceland Basin from Icelandic runoff and possibly also the East Icelandic Current. The dashed blue arrow indicates the proposed anomalous flow bringing water from the Icelandic shelf/slope region into the western Iceland Basin. Dark blue squares indicate the locations of four OSNAP moorings. Cyan line indicates the north-western part of the Extended Ellett Line (EEL). RR: Reykjanes Ridge. IFR: Iceland-Faroe Ridge. RHP: Rockall-Hatton Plateau. RT: Rockall Trough. FSC: Faroe-Shetland Channel. **(b)** Annually averaged salinity for each of the three warm-water branches (brown lines) and the reference site (RT, black line) since 1970 from the ICES IROC database (https://ocean.ices.dk/iroc/). The shaded area indicates the 2015–2018 period.

The post-2014 freshening seems to have been most pronounced in the Iceland Basin (Fig. 1a) which was characterised by surface waters that were fresher than previously seen in 120 years of surface observations and nearly 50 years of subsurface observations (Holliday et al, 2020). The event is thought mainly to be a consequence of changes in the regional winter wind stress that caused a diversion of Arctic-origin low salinity water from the western boundary into the eastern basins via the North Atlantic Current. Additionally, the same wind stress pattern stimulated a zonal shift of the subpolar front, bringing fresher water into the Iceland Basin and Rockall Trough.

In this manuscript we examine the role of an additional freshwater source that contributed to the surface signal, and may explain why the freshening event was most extreme in the Iceland Basin (Holliday et al., 2020). This additional freshwater source is a low-salinity water mass found over the south-Icelandic shelf/slope region, often termed the Icelandic Coastal Current (Ólafsson et al., 2008; Logemann et al., 2013). The low salinity of this water mass mainly derives from Icelandic runoff, which is estimated to carry more than 5000 m$^3$ s$^{-1}$ (5 mSv) of freshwater into the ocean on average. The runoff has a high seasonal variation with maximum around June (Gröndal, 2004; Jónsdóttir, 2010).



In addition to this, low-salinity water from the East Icelandic Current (Jónsson, 2007; Macrander et al., 2014) may also contribute a freshwater input. Most of this water is generally considered to continue into the Norwegian Sea (Stefánsson, 1962; Perkins et al., 1998; Valdimarsson and Malmberg, 1999; Jónsson, 2007; Logemann et al., 2013). Some of

the water over the shelf east of Iceland may round the corner and enter the shelf/slope region south of Iceland (blue arrow in Fig. 1a), but the amount is not well quantified.

Whether the runoff from Iceland is supplemented by low-salinity waters from the East Icelandic Current or not, there is a considerable amount of freshwater entering the shelf/slope region south of Iceland. Typically, this water is considered either to join the Atlantic water flow towards the Iceland-Faroe Ridge into the Norwegian Sea or towards the

Denmark Strait into the Iceland Sea (Valdimarsson and Malmberg, 1999; Logemann et al., 2013).

In some periods this water may, however, take an alternative "south-western" route, flowing from the shelf/slope region south of Iceland south-westwards along the eastern slope of the Reykjanes Ridge into the western parts of the Iceland Basin (dashed blue arrow in Fig. 1a). The existence of this route is well known and was documented by Valdimarsson and Malmberg (1999) based on the paths of drifters deployed south of Iceland, but it has generally been overlooked as a potential

source of freshwater input to the SPNA.

In this study, we use satellite altimetry to show that the upper-layer circulation of the Iceland Basin after 2014 was favourable for transporting low-salinity water from the Icelandic shelf/slope region along this route. We use a variety of observational data sets to examine the effect of this "Icelandic freshwater source" on the salinity of the Iceland Basin and the three current branches of warm water flow into the Arctic Mediterranean. We consider the effect of the very fresh surface

water on the stratification and depth of winter convection in the Iceland Basin and analyse the causal effect of wind-forcing on the anomalous circulation.

## 2 Material and methods

### 2.1 Satellite altimetry

Altimetry data were selected from the global gridded ($0.25° \times 0.25°$) fields representing Mean Dynamic Topography (MDT)

and daily averaged Sea Level Anomaly (SLA) for 9629 days from 1 January 1993 to 15 May 2019 available from Copernicus Marine Environment Monitoring Service (CMEMS) (https://resources.marine.copernicus.eu/?option=com_csw&view=details&product_id=SEALEVEL_GLO_PHY_L4_REP_OBSERVATIONS_008_047).

### 2.2 Satellite-tracked drifters

Quality controlled data, interpolated to 6-hour intervals, from satellite-tracked drifter buoys in the area (0°–30° W, 50° N–65° N) were downloaded from NOAA's Atlantic Oceanographic and Meteorological Laboratory (AOML)



(http://www.aoml.noaa.gov/envids/gld/dirkrig/parttrk_spatial_temporal.php). The drifters are drogued at 15 m depth and only data with the drogue attached are used here. The drifter data have been averaged to daily positions before use.

### 2.3 Hydrographic data

Hydrographic observations from three different sources are used. Salinity values are reported in practical salinity units (psu).

### 2.3.1 Extended Ellett Line

We use salinity observations from eleven cruises along the Extended Ellett Line between Iceland and the Rockall-Hatton Plateau. The data have been interpolated horizontally onto an equidistant grid.

### 2.3.2 OSNAP moorings

Four OSNAP moorings in the Iceland Basin (M1, M2, M3, M4, Fig. 1a) measured temperature and salinity at several depths from July 2014 to July 2018 using Microcats. We use the salinity values from the top six measurement depths: 50 m, 100 m, 200 m, 350 m, 500 m, and 700 m. The salinity was measured every half hour although instrumental problems and quality assurance has introduced some gaps. At sites M1 and M2, the data from 50 m depth only extend to July 2016. We also use values for potential density at these moorings.

### 2.3.3 Icelandic standard sections

Salinity data from three standard sections south of Iceland (Supplementary Table S1) are used: section SB (Selvogsbanki), section IH (Ingólfshöfði), and section ST (Stokksnes). The data include all cruises in the 1993–2019 period, for which all stations at each section were occupied (87 cruises at SB, 86 cruises at IH, and 94 cruises at ST).

### 2.4 Meteorological data

Monthly averaged surface air pressure and net precipitation (total precipitation minus evaporation) were downloaded from Copernicus Climate Change Service (C3S) (2017): ERA5: Fifth generation of ECMWF atmospheric reanalyses of the global climate. Copernicus Climate Change Service Climate Data Store (CDS), 30. September 2020.

### 2.5 Statistical methods

Statistical significance of correlation coefficients is corrected for serial correlation by the modified Chelton method according to Pyper and Peterman (1998). Empirical Orthogonal Function (EOF) analysis follows Preisendorfer (1988).




## 3 Results

### 3.1 Satellite altimetry

Since the slope of the sea surface is linked to the geostrophic flow of the upper layers, satellite altimetry may be used to answer two important questions: 1) what is the typical flow pattern through the upper parts of the Iceland Basin? and 2) were

there any variations to this typical flow during the extreme freshening event that may help explain the event?

To illustrate the typical flow patterns in the region, we added the Mean Dynamic Topography (MDT) to the average values of the Sea Level Anomaly (SLA) for the 1993–2018 period (Fig. 2). This makes the result independent of any assumptions used in generating the MDT although the result will still depend on the accuracies of the altimetry and of the geoid.

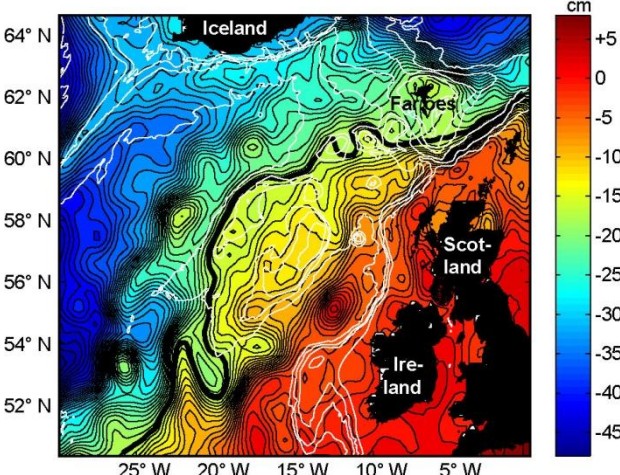


**Figure 2.** Average sea surface height between 1 Jan 1993 and 31 Dec 2018 generated as the sum of the MDT and SLA values averaged over the period. Sea surface height is indicated by background colours and the black lines with the thick black line indicating the boundary between waters passing north of and south of the Faroes according to the altimetry. White lines are bottom contours.

In this framework, there is an anticlockwise circulation in the north-western Iceland Basin, which is consistent with

vessel-mounted ADCP observations (Chafik et al., 2014). Over the rest of the basin, the meridional component of the near-surface flow is, however, towards the north and Fig. 2 has no indication of an average flow from the Icelandic shelf/slope region into the central Iceland Basin.

The average flow pattern indicated by Fig. 2 will, however, be distorted by synoptic and mesoscale variability and possibly also by more persistent climate variations. To investigate temporal variations, an EOF analysis was performed on

the SLA data that were condensed into non-overlapping 28-day averages to reduce synoptic and mesoscale variability. To enhance variations of sea surface slope, a new set of "modified" SLA values was generated by subtracting the spatial average for each time step (see Supplementary methods).



The first EOF mode of the modified SLA data (Fig. 3) explains 25% of the variance. For annual (instead of 28-day) averages, the first EOF mode of the modified SLA looks very similar (Supplementary Fig. S4) and explains 56% of the variance. Thus this mode represents the dominant variability of geostrophic surface velocity on long time scales, and this mode did exhibit anomalous behaviour during the freshening event.

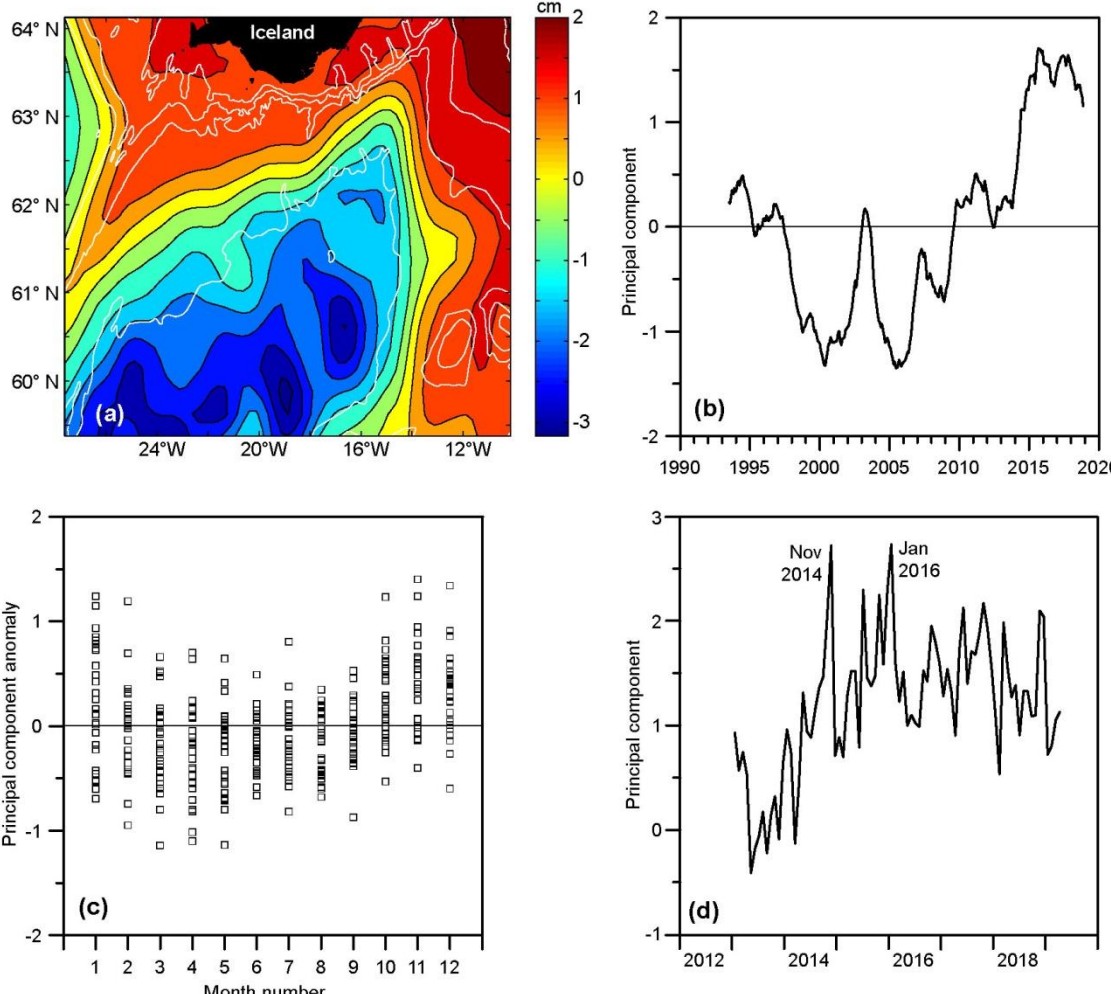

**Figure 3.** The first EOF mode of 28-day averaged modified (Supplementary methods) sea level height anomaly from satellite altimetry. **(a)** Spatial structure of the mode with depth contours (white lines) for 200 m, 500 m, 1000 m, and 2000 m. **(b)** Low-passed (running mean over thirteen 28-day periods, equivalent to 364 days) principal component. **(c)** Seasonal variation of the principal component anomaly defined as the deviation from the low-passed principal component. Each square represents the principal component anomaly for one 28-day period. **(d)** Detailed development of the principal component prior to and during the freshening event.

In the following, the principal component of this mode for 28-day averaged modified SLA will be denoted just "the principal component" and its temporal variations are illustrated in Fig. 3. During the first part of the altimetry period, the low-passed principal component (Fig. 3b) was close to zero or negative, representing enhanced clockwise circulation in the



Iceland Basin. After 2014, however, it was primarily positive, representing enhanced anticlockwise circulation that might bring water from the shelf/slope region south of Iceland in a southwest-ward direction towards the western Iceland Basin.

In addition to the long-term variation, the principal component exhibits pronounced seasonality with enhanced anticlockwise circulation in late autumn to early winter (Fig. 3c). The combination of long-term and seasonal variations contributed to making the two most extreme positive values for the principal component during the whole altimetry period to occur around the time of the extreme freshening event (Fig. 3d).

### 3.2 Satellite-tracked drifters

With drogues at 15 m depth, the drifter tracks present an alternative (to altimetry) method to study the upper layer flow and check whether they support a pathway from the Icelandic shelf/slope region into the central Iceland Basin (dashed blue arrow in Fig. 1a). As a whole, the drifter tracks (Supplementary Fig. S5) are quite consistent with the typical flow pattern based on altimetry (Fig. 2). The drifter tracks do, however, also indicate temporal variations of the flow field, as reported by Valdimarsson and Malmberg (1999).

To study this in more detail, we selected all drifters that entered the shaded region labelled "ISS" on Fig. 4 and plotted their tracks after entering the region. Out of the 88 drifters that entered this region, 51 left the Iceland Basin eastwards into or towards the Norwegian Sea (red tracks in Fig. 4), whereas 20 drifters exited westwards, all but one continuing towards the Denmark Strait (blue tracks). The remaining 17 drifters (cyan tracks) stopped transmitting position or lost their drogues within the Iceland Basin (including the Icelandic shelf/slope region).

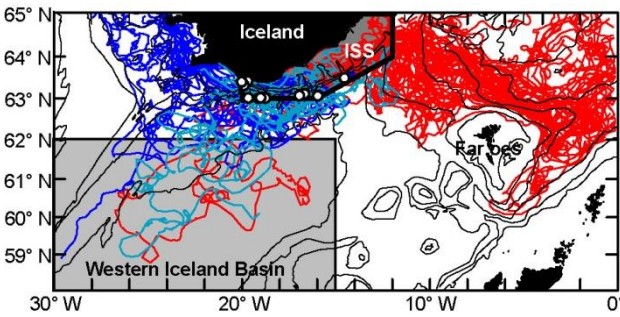

**Figure 4.** Tracks of all the drifters that were deployed in or entered the ISS-region (Icelandic Shelf/Slope), showing their paths from the time that they entered the region. Different colours indicate whether the drifter exited the region towards the east (red) or west (dark blue), or stopped functioning within the region (cyan). White circles indicate the entering points into the ISS-region for 11 drifters that later entered the shaded region labelled "Western Iceland Basin".

After entering the ISS-region, most of the drifters remained close to Iceland while in the region. Some of the drifters continued in a south-westward direction into more oceanic areas, however, and 11 of these passed south of 62° N into the shaded region labelled "Western Iceland Basin" in Fig. 4 (Supplementary Table S2). As shown by the white circles in the figure, most of these 11 drifters had entered the Icelandic shelf/slope region around 20° W, but some of them entered it farther towards the east.


The tracks of these 11 drifters towards the southwest follow the streamlines indicated by the EOF mode in Fig. 3a and demonstrate that this mode is not only a deviation from the average flow field, but may dominate the flow
(Supplementary Fig. S2). The travel of these drifters from the time that they entered the Icelandic shelf/slope region until they entered the box labelled Western Iceland Basin typically lasted a couple of months (Supplementary Table S2).

With only 11 drifters taking this path, it would be hard to determine temporal variations. Instead, we have checked whether there is any consistency between drifter movement and the SLA data from altimetry. To do this, the positions of the drifters after they had entered the ISS-region in Fig. 4 were determined at the start and end of each 28-day period used in the
EOF analysis as long as the drifters stayed within the Iceland Basin. The drift during each of these periods (end longitude minus start longitude) was then correlated with the principal component during the period. The correlation coefficient was –0.37 and significantly different from zero (p<0.001). Considering the difficulty in comparing Lagrangian and Eulerian data, this indicates consistency between the drifter and altimetry data.

## 3.3 Salinity observations

### 3.3.1 Salinity on the Extended Ellett Line (EEL)

The EEL hydrography sections exhibit a highly variable salinity structure across the Iceland Basin as exemplified in Supplementary Fig. S6. Figure 5 shows the variation of the top-200 m average salinity across the basin in each of the eleven years. Some years (black curves in Fig. 5), the salinity increases almost linearly from the slope region south of Iceland to the Rockall-Hatton Plateau. In other years (reddish curves), the near-surface waters in the central basin are much fresher than the
two sides of the basin, whereas there are also intermediate years (cyan curves).

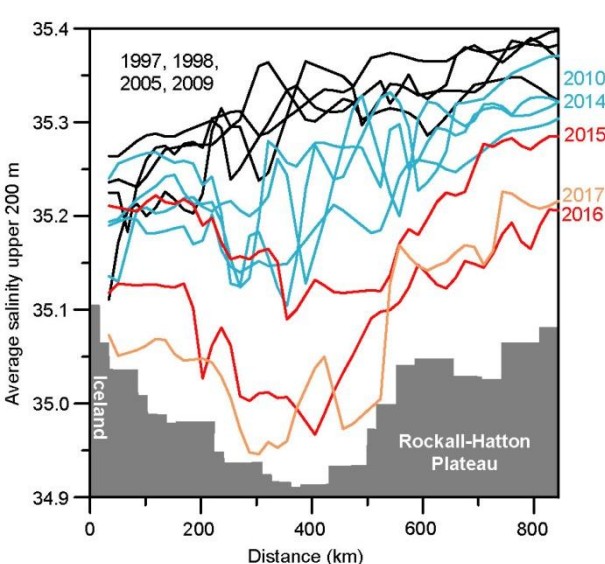

**Figure 5.** Salinity averaged over the top 200 m plotted against the distance from Iceland from eleven EEL cruises. The shaded area indicates bottom topography.

The three most extreme years in Fig. 5 (2015–2017) are characterized by low salinity across the whole of the
Iceland Basin and onto the Rockall-Hatton Plateau, but especially in the middle of the basin where a low-salinity water mass
is squeezed between two cores of water with higher salinities.

### 3.3.2 Salinity at the OSNAP moorings

**Figure 6.** Monthly averaged salinity at depths 50 m (dark blue), 100 m (cyan), 200 m (red), and 350 m (brown) at the four OSNAP
moorings. Only months, for which at least 28 days had acceptable data are included at each depth level.

The salinity at all four of the OSNAP moorings exhibited decreasing trends for the uppermost 350 m through the
four-year deployment period (Fig. 6). At mooring sites M3 and M4, the decrease was rapid during the first half of the period,
followed by more stable or increasing salinities. At sites M1 and M2, a similar development was seen for the 100 m level





(and for 200 m at M2), whereas the salinity at 350 m (and at 200 m for M1) had decreasing trends throughout the four years.
This could indicate a lagged response to the freshening signal coming from the western boundary when taking into account the anticlockwise circulation over the eastern flank of the Reykjanes Ridge (Fig. 2). This circulation pattern would also explain the zonal variation of the average salinity (Supplementary Fig. S7b).

For this study, the most interesting aspect of the OSNAP data is the information on near-surface salinity and its vertical variation since this might be related to a local near-surface freshwater source. Averaged over the whole deployment
period, the salinity increased with depth in the uppermost layer with maximum salinity located somewhere between 200 m and 500 m (Supplementary Fig. S7a). This average picture masks, however, considerable variation. At site M2, the top 350 m look homogeneous in salinity at the beginning and end of the four-year deployment period, but permanently stratified in between (Fig. 6). At the other mooring sites, periods with homogeneous salinity are interrupted by periods with strong stratification lasting several months. This indication of near-surface freshwater pulses will be further discussed in Sect. 4.1.

### 3.3.3 Salinity and freshwater at Icelandic standard sections

The three Icelandic standard sections south of Iceland are shown in Fig. 7a. On each section, the standard stations are labelled from 1 up to 5, starting from the station closest to the coast. The average near-surface salinities (red lines on Fig. 7c) clearly show the low-salinity characteristics of the coastal current and the freshwater content on the sections is illustrated in Fig. 7c.

The concept of freshwater content is often criticized for its arbitrariness (e.g., Schauer and Losch, 2019). Here, we are, however, considering sea waters that have recently received pure freshwater, primarily as runoff from Iceland. The freshwater content of these waters should therefore be well-defined as long as the salinity of the oceanic water, with which the freshwater has been mixed, is used to determine the mixing ratio.

In this case, this "reference salinity" should represent the Atlantic water approaching the Icelandic shelf/slope
region and it should be expected to vary considerably with time. From the typical flow field (Fig. 2) this water passes through the outermost part of the ST section and this is also where the highest near-surface salinities are found on average (red curves in Fig. 7c). To get a time series for the salinity of the Atlantic water mixing with the freshwater, we therefore use the salinity observations at station ST5. For each cruise, the 50 m thick layer with highest salinity at this station was identified, and annual averages of its salinity are used to represent Atlantic water salinity (Fig. 7b).

Using this time series as reference salinity for each year separately allows determination of a "freshwater thickness" for each station from each cruise by extrapolating the salinity profile to the surface and to the bottom and integrating over the whole water column or over the uppermost 200 m for the deep stations. On average, the freshwater thickness defined in this way (cyan bars in Fig. 7c) is highest at the innermost stations even though they are the shallowest (Supplementary Table S1). Interpolating linearly between the stations, the freshwater areas on the sections were then determined for each cruise by
horizontal integration.





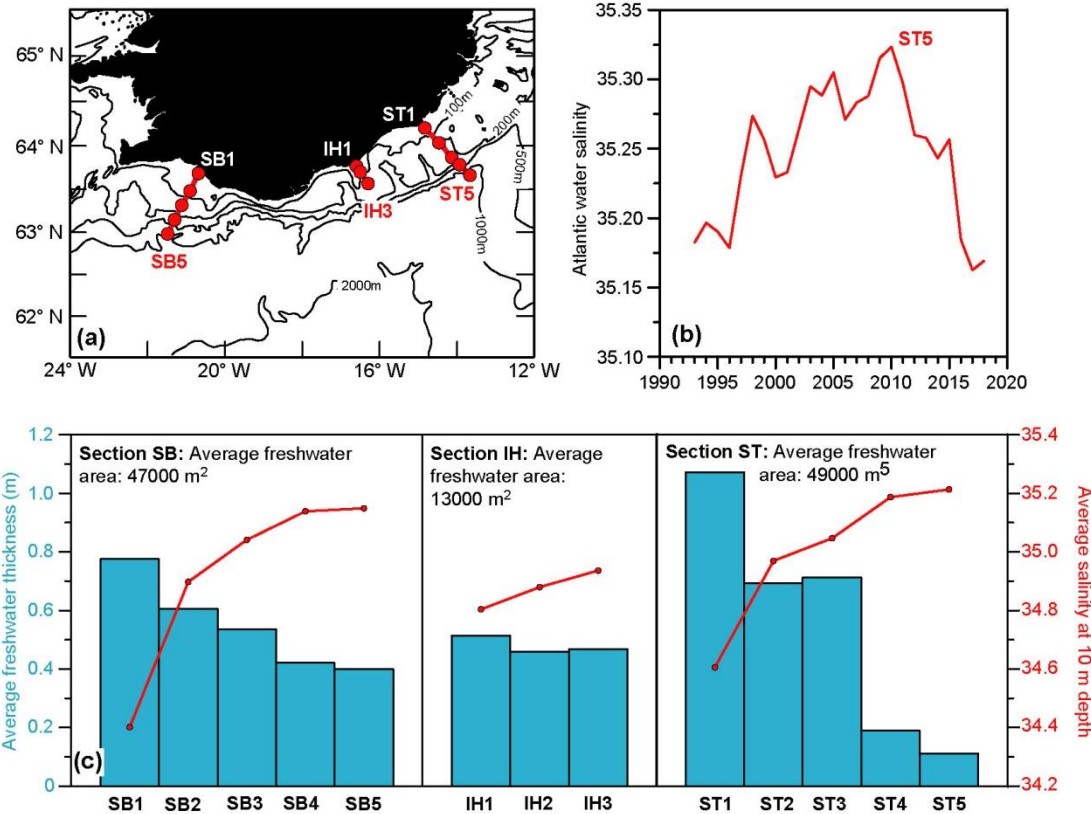

**Figure 7.** Salinity and freshwater over the south-Icelandic shelf/slope region. **(a)** The three standard sections with a total of 13 standard stations (red circles). **(b)** Annually averaged maximum salinity in a 50 m layer at station ST5 (red curve). **(c)** Average salinity at 10 m depth (red circles and lines, right scale) and average freshwater thickness down to the bottom or 200 m (cyan bars, left scale) at each
standard station.

The freshwater area for each of the sections varied considerably between cruises (Supplementary Fig. S8). For the two standard sections that cover the whole shelf width (SB and ST), the average freshwater area was close to 50000 m$^2$ (Fig. 7c). Putting this number into perspective is difficult without further information, but one may note that a constant current of 10 cm s$^{-1}$ through each of these two sections would give a freshwater transport of 5 mSv, i.e. equal to the freshwater runoff

from Iceland.

Section IH has a considerably lower average freshwater area. Partly, because it does not cover the whole width of the shelf (Fig. 7a), but it also seems to be located in a divergence zone between eastward- and westward-flowing waters (Fig. 4).



## 4 Discussion

### 280   4.1 The origin of the post-2014 freshening event in the Iceland Basin

Holliday et al. (2020) have discussed different processes that may have contributed to the extreme freshening of the Iceland Basin and this discussion will not be repeated here. Instead, we focus on the freshwater contribution from a source that was not discussed by Holliday et al. (2020): the shelf/slope waters south of Iceland, fed by Icelandic runoff and possibly the East Icelandic Current. We term this "the Icelandic freshwater source".

285        As demonstrated by the drifter tracks (Fig. 4), these waters will normally flow into the Nordic Seas, either eastwards across the Iceland-Faroe Ridge or north-westwards through the Denmark Strait. In some periods, however, the drifters indicate a circulation pattern similar to the one associated with the first EOF mode of modified SLA (Fig. 3), which can bring them, first into the western, and then into the central parts of the Iceland Basin. This anomalous circulation connects the central basin with the Icelandic shelf/slope region and constitutes a freshwater source which will be surface-

intensified until homogenized by vertical mixing and convection.

       Although salinities in the eastern SPNA started to decrease already around 2010 (Fig. 1b), the extreme freshening event in the Iceland Basin seems to have been initiated in late 2014 and became much more pronounced in 2015 (Fig. 5 and Fig. 6). This fits well with the temporal development of the circulation pattern associated with the first EOF mode of modified SLA (Fig. 3), which had an extremely high positive value for the principal component in November 2014 and

remained high at least until the end of 2018 (Fig. 3d).

       Thus, the surface-intensified freshening pulses indicated by the OSNAP data (Fig. 6) might be explained by this anomalous circulation pattern bringing low-salinity water from the Icelandic shelf/slope region into more central parts of the Iceland Basin. As indicated by the drifter data (Supplementary Table S2), this passage would only require a few months. If, alternatively, the low salinity near the surface was caused by waters that originated off the western boundary a few years

earlier (Holliday et al., 2020), then we would not expect the short-term variations seen in the salinity stratification (Fig. 6).

       To get an impression of the amount of freshwater needed to generate the near-surface salinity decreases in Fig. 6, we can define a freshwater thickness for each of the OSNAP moorings as the thickness of a freshwater layer that would be needed to mix (instantaneously) with the water at 350 m depth at the mooring to give the observed average salinity of the uppermost 350 m. Time series of this parameter (Fig. 8) show rapid development of the near-surface freshening. The

anomalous circulation pattern would therefore only need to persist for a few months to build up each of the freshwater pulses indicated by Fig. 8. The most pronounced freshwater pulse during the OSNAP deployment was initiated in 2015. It seems to have arrived at M2 in April 2015, followed by M3, then M1, and finally M4 in late July. For low-salinity water from the Icelandic shelf/slope region to generate these signals, the anomalous circulation pattern of Fig. 3a would need to prevail during less than six months; followed by more normal flow (Fig. 2).




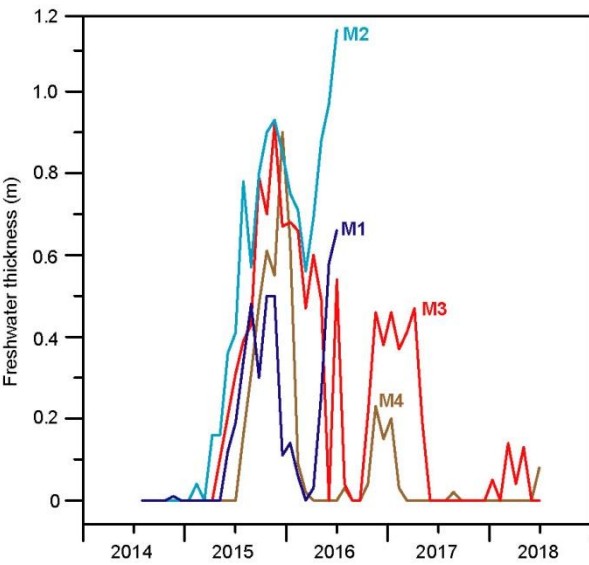


**Figure 8.** Freshwater thickness at the OSNAP moorings. The coloured lines show 28-day averaged freshwater thickness of the uppermost 350 m layer at each of the four OSNAP moorings (Fig. 1a). For days with the average salinity of the layer exceeding the salinity at 350 m, freshwater thickness is set to zero.

Such a scenario may also explain the near-surface salinity variations across the basin observed on the Extended

Ellett Line (EEL). On this section, the most extreme fresh years (2015, 2016, and 2017) were characterized by the near-surface parts of the central basin being considerably fresher than the regions on both sides (i.e., close to Iceland and close to the Rockall-Hatton Plateau) (reddish curves in Fig. 5). This is in contrast to other years where the salinity varied more linearly (black curves) or more irregularly (cyan curves) across the basin. The EEL observations in 2015–2017 were obtained in May–June, at a time of year when the principal component is usually close to its minimum (Fig. 3c). Thus the

low-salinity waters in the central basin for these years could derive from the Icelandic freshwater source if they were transported into the oceanic parts of the basin west of the EEL section at an earlier time. After this, the normal north-eastward flow (Fig. 2) would distribute these low-salinity waters through the central parts of the basin.

During the most intensive freshening of the central Iceland Basin from late-2014 to mid-2016, it also appears that the shelf/slope region south of Iceland was almost drained of freshwater. Certainly, the freshwater areas of all the Icelandic

standard sections in this region decreased substantially (Fig. 9) and both section ST and SB experienced their lowest freshwater area of the whole 27-year time series during the first half of 2016. This drainage may be explained by the anomalous transport of coastal water into the oceanic parts of the Iceland Basin during this period.

From the above, there are several arguments supporting a freshwater input from the Icelandic freshwater source. The question remains, however, whether this source can supply sufficient amounts of freshwater to explain the observed

freshening. From Fig. 5, the salinity of the upper 200 m was ≈0.1 lower in the central basin than on the two sides. If the water in the central basin is seen as a mixture of oceanic water from both sides and pure freshwater added on the top, then the thickness of this freshwater layer would have to be around 0.5 m. For the OSNAP data (Fig. 8), an average value of 0.5



m also seems realistic. From Fig. 5, the width of this low-salinity layer is around 300 km. If we use a similar value for the extent in the perpendicular (southwest-northeast) direction, the total area is around $10^{11}$ m$^2$, which implies a freshwater

volume of $5 \cdot 10^{10}$ m$^3$. With a steady freshwater supply over a year, this is equivalent to 1.6 mSv.

As mentioned in Sect. 1, the average runoff from Iceland is more than 5 mSv. Thus, only around one third of that is needed to explain the extra upper-layer freshening during 2015–2017. The runoff is distributed all around the Icelandic coasts, but the fraction entering south Icelandic waters is more than one third (Gröndal, 2004). In addition to Icelandic runoff, some low-salinity water from the East Icelandic Current may have rounded the southeast corner of Iceland and

contributed to the freshening, but we have no evidence that can confirm or reject that. It is not clear, however, that any water from the East Icelandic Current is needed. From the calculations above, runoff from Iceland should be sufficient to supply the needed freshwater.

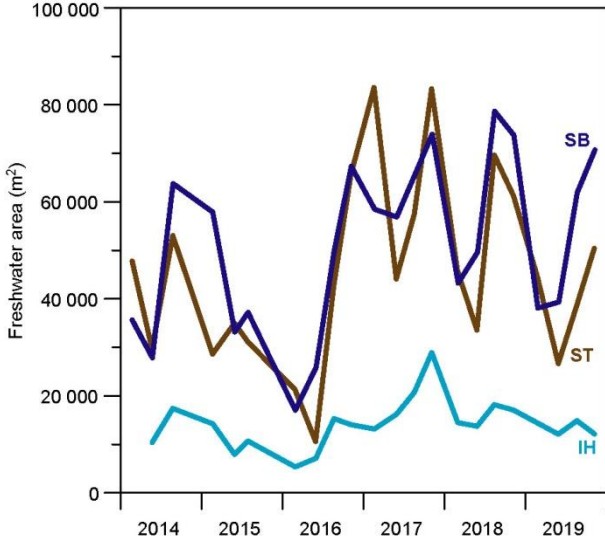

**Figure 9.** The thick coloured curves show the freshwater areas at the three standard sections south of Iceland (Fig. 7) for cruises in the
2014–2019 period.

An alternative explanation of the near-surface freshening might be excess precipitation. In their study, Holliday et al. (2020) concluded that increased net precipitation only played a minor role in the freshening. For our short-term, surface-intensified additional freshening signal, enhanced net precipitation might conceivably be more important. The net precipitation over an area at the western entrance to the basin covering the OSNAP moorings had an increasing trend from

2010 to 2018 (Supplementary Fig. S9). It did not, however, indicate exceptional conditions during the surface freshening events (Fig. 8) and was too small to supply the necessary amount of freshwater.

Thus, there are several indications that the Icelandic freshwater source made a significant contribution to the near-surface freshening in the Iceland Basin after 2014. A weakness with this interpretation is that the pattern in Fig. 3a represents the deviation from the mean flow; not the flow itself. Ideally, this could be rectified by calculating the Absolute Dynamic

Topography (ADT) for the period when the principal component peaks by adding the SLA for that period to the Mean





Dynamic Topography (MDT). That requires, however, that the MDT is sufficiently accurate, which is not guaranteed on such small scales near land and steep topography (Rio et al., 2011). For the shelf/slope region south of Iceland, the flow pattern associated with the average SLA for this period actually looks more realistic than the pattern associated with the average ADT (Supplementary Fig. S2 and discussion in Supplementary methods).

Irrespective of this uncertainty, the drifter data clearly show that water does flow from the Icelandic shelf/slope region into the western Iceland Basin (Fig. 4) and that this did occur also during the extreme freshening period in 2015 (Supplementary Table S2). A priori, the drifters only represent the Ekman layer, but drifter velocities are significantly correlated with the principal component, which represents a more extensive water mass (Sect. 3.2). Combining this with the evidence listed above, we find it most likely that the Icelandic freshwater source was the primary cause of the near-surface

freshening events in the 2015–2017 period (Fig. 8). Arriving on top of the more extensive freshening discussed by Holliday et al. (2020), these events made the surface freshening in the Iceland Basin so extreme.

### 4.2 The freshening of warm-water inflow to the Nordic Seas

As mentioned in Sect. 1, one of the consequences of the freshening eastern subpolar North Atlantic was a freshening of all the branches of warm-water inflow to the Nordic Seas (brown arrows in Fig. 1). Here, the traditional term "Atlantic inflow"

(e.g., Hansen and Østerhus, 2000) will be used to denote this flow. High salinities of the Atlantic inflow branches are preconditions for sufficient surface density to allow thermohaline ventilation. It is therefore of interest to clarify how much of the Atlantic inflow freshening derived from the Icelandic freshwater source discussed in this manuscript rather than the sources discussed by Holliday et al. (2020).

Using 2014 as a baseline, Table 1 lists how salinity changed from that year to the following years for the three

Atlantic inflow branches as well as at the reference site in the Rockall Trough (Fig. 1). In the table, the values are listed with two decimals, but the accuracy varies between the series and most of them probably do not reach an accuracy of 0.01, when aliasing and under-sampling are taken into account. Thus, the values should be interpreted with caution, but some rough estimates can be made.

**Table 1.** Salinity changes from 2014 to the following years at Rockall Trough (RT), and at the monitoring sites for the three Atlantic inflow branches (Fig. 1). All values are derived from the data in the ICES IROC database (https://ocean.ices.dk/iroc/).

|           | RT    | S-br. | F-br. | I-br. |
|-----------|-------|-------|-------|-------|
| 2014–2015: | +0.02 | −0.02 | +0.02 | +0.06 |
| 2014–2016: | −0.07 | −0.06 | −0.02 | +0.02 |
| 2014–2017: | −0.07 | −0.09 | −0.09 | −0.04 |
| 2014–2018: | −0.06 | −0.09 | −0.09 | −0.09 |





From the typical flow field (Fig. 2) and the enhanced anticlockwise circulation during the freshening event (Fig. 3), water from the central Iceland Basin would not be expected to influence the western Rockall Trough appreciably. Thus, the
freshening at site RT in Table 1 is likely explained by the mechanisms discussed by Holliday et al. (2020) with no contribution from the Icelandic freshwater source. This justifies the use of RT as a reference site.

For the S-branch, Table 1 shows more freshening than at site RT. The monitoring for this branch is located on the Shetland shelf and the influence from the Iceland Basin should be minimal, especially with the enhanced anticlockwise circulation (Fig. 3). Much of the water at this location also derives from the Continental Slope Current (Booth and Ellett,
1983), which may be affected by quite different processes. Thus, the freshening of the S-branch after 2014 is not likely to have been caused by the Icelandic freshwater source to any large extent.

For the F-branch, more or less all the water is expected to pass through the Iceland Basin under normal conditions (Fig. 2 and Supplementary Fig. S5). The average volume transport of the F-branch is 3.8 Sv (Hansen et al., 2015), which is equivalent to $1.2 \cdot 10^{14}$ m$^3$ per year. Compared to this, one pulse of $5 \cdot 10^{10}$ m$^3$ of freshwater from the Icelandic source (Sect
4.1) would lower the annually averaged salinity by ≈0.015. With the sluggish flow through the central parts of the basin (Fig. 2), some delay would be expected. Freshwater input from the Icelandic source may therefore be the main reason that the salinity decrease from 2014 to 2017 was stronger for the F-branch than for the RT site. For the total freshening of the F-branch during this period (Table 1), the Icelandic freshwater source only contributed at most 20%, however.

Much of the water feeding the third Atlantic inflow branch, the I-branch, comes from the Irminger Basin, but some
of it passes through the Iceland Basin (Fig. 2) and the anomalous circulation associated with the first EOF mode of the modified SLA data also affected the eastern boundary of the Irminger Basin (Fig. 3). Thus, some of the freshening observed for the I-branch may come from the Icelandic freshwater source, but the salinity of this branch is highly variable and its freshening seems delayed compared to the other branches.

Thus, it is difficult to estimate how much the Icelandic freshwater source contributed to the I-branch. When taking
into account its volume transport and that of the other Atlantic inflow branches (Østerhus et al., 2019), we can in any case conclude that the freshening of the total Atlantic inflow to the Nordic Seas from 2014 to 2018 (all branches) owed at most 10% to the Icelandic freshwater source.

### 4.3 Stratification and winter convection

If not counteracted by temperature, surface-intensified freshening will increase stratification and may act to reduce the depth
of winter convection that brings up nutrients from deeper layers and improves conditions for primary production. Surface-intensified freshening would be expected to reduce the convection depth, but determining whether that has been the case is not an easy task. With a given hydrographic profile (e.g., from CTD or ARGO float) the instantaneous convection depth at the location may be estimated as the maximum depth for which the potential density equals the surface density within the uncertainty interval. A high day-to-day variability may be expected, however (de Jong et al., 2012). For unambiguous


determination of maximum convection depth at a specific site during one winter, hydrographic profiles would therefore be required at the site at least once a day during the winter.

Such an ideal data set is not available for any location in the central Iceland Basin and the OSNAP data are not ideal. With half-hourly measurements, they may be considered temporally continuous, but they only measure at a few discrete depths. A special deficiency of the OSNAP data is the lack of surface measurements. This does not necessarily

imply that no information on convection depth can be gained from this data set as illustrated in Fig. 10. In this figure, there is seen to be a difference in potential density between 50 m depth and 350 m depth during most days in the period. Thus the convection depth is less than 350 m during those days. In early 2018, the two densities converge, however, and remain almost equal for some time. Since the density at 50 m depth is mainly governed by heating and cooling from above, it also seems likely that this equality extends to the surface, indicating convection down to at least 350 m during these days.

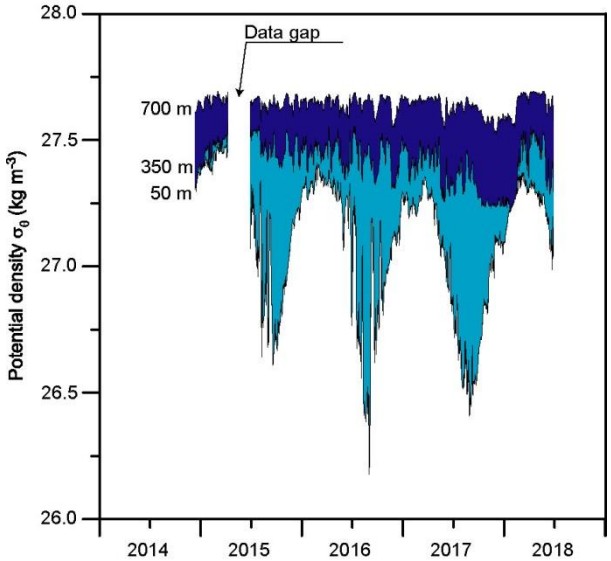


**Figure 10.** Stratification at the OSNAP mooring M3 (Fig. 1a). Daily averaged potential density ($\sigma_\theta$) at three depth levels at mooring site M3. The coloured areas indicate the differences in potential density between 50 m and 350 m (cyan) and between 350 m and 700 m (dark blue).

In the following, we will adopt this interpretation and assume that equality between potential density at 50 m and a

deeper level may be used to indicate convection down to that level at the time. Instead of daily averages, we will use the individual (half-hourly) measurements. To prevent false signals from spikes in the data, convection down to a given level is defined to occur on a given day if the difference in potential density between the level and 50 m depth is less than 0.01 kg m$^{-3}$ for at least 12 of the 48 measurements during the day. By this criterion, none of the four moorings indicated convection down to the instruments at 700 m depth during the four winters, but three of them experienced days with convection down to

500 m some years (Supplementary Table S3). In cases where data from 50 m depth were missing, the potential density at 100 m depth was used instead (Supplementary Table S4). Although there are differences in details, these two criteria



(Supplementary Table S3 and Supplementary Table S4) agree on the maximum depth of convection each winter, which is summarized in Table 2.

**Table 2.** Depths (m) of the instruments, down to which convection was indicated at each of the OSNAP moorings during each of the four winters. The ≥ sign for M3 is because data from the instrument at 500 m depth were missing for the 2014–2015 winter.

| Winter | M1 | M2 | M3 | M4 |
|---|---|---|---|---|
| 2014–2015: | 500 | 500 | ≥350 | 500 |
| 2015–2016: | 500 | 350 | 200 | 350 |
| 2016–2017: | 500 | 200 | 200 | 350 |
| 2017–2018: | 500 | 350 | 350 | 500 |

From Table 2, it appears that the two winters most affected by near-surface freshening (2015–2016 and 2016–2017) had reduced convection at all of the moorings except M1. It seems likely that this reduction in convection depth was caused
by the extra freshwater input to the near-surface waters from the Icelandic freshwater source. Although the post-2014 surface freshening probably was anomalous, the drifter data demonstrate that flow from the Icelandic shelf/slope region into the Western Iceland Basin also occurred prior to this (Supplementary Table S2). From the seasonal variation of the principal component (Fig. 3c), freshwater input from Icelandic runoff is furthermore most likely to occur around or shortly before the most intense atmospheric cooling.

Thus, reduced winter convection and reduced nutrient renewal of the euphotic zone in the central Iceland Basin may be a more general phenomenon, which could help explain why this region seems to be comparatively oligotrophic (Pacariz et al., 2016; Hátún et al., accepted). Further studies are, however, needed to assess the validity of this hypothesis.

## 4.4 Wind forcing

The question arises what the cause was of the anomalous circulation in the Iceland Basin after 2014 (Fig. 3b). In their study,
Holliday et al. (2020) noted that the winter (DJFM) wind stress curl over the basin was anomalously high in 2014–2016 (their Fig. 9). It seems likely that this could increase the strength of the circulation mode that transported low-salinity waters from the Icelandic shelf/slope region into the central Iceland Basin (Fig. 3). To investigate this, we have decomposed the ERA5 surface air pressure field into the same spatial modes as the EOF modes of modified altimetry SLA (Supplementary Methods, Eq. (S4)), where the weighting function, termed $p_n(t)$, associated with each mode, $n$, indicates how strongly that
mode is represented in the pressure field at any given time.

When the weighting function for the first mode $p_1(t)$ is high, the isobars of the surface air pressure tend to be aligned with the contour lines of modified SLA associated with this mode (Fig. 3a). The geostrophic wind will then tend to parallel the geostrophic surface velocity anomaly associated with the mode, thus reinforcing it.





This "wind-forcing parameter" therefore ought to reflect the ability of the wind to generate the circulation associated with this mode. To check this, we have correlated $p_1(t)$ with the principal component (Fig. 3). For monthly averages, the correlation coefficient was 0.27, which is significantly greater than zero at the 99.9% level ($p<0.001$). For annual averages, the correlation coefficient increased to 0.42, which is significantly greater than zero at the 95% level ($p<0.05$).

Thus, $p_1(t)$ may be considered a forcing parameter, which to some extent represents the capacity of the wind field to excite the circulation associated with the first mode of the modified SLA data (Fig. 3a). As shown in Fig. 11, this forcing factor reached its maximum in 2014, a few months before the circulation anomaly associated with the first SLA-EOF mode culminated (Fig. 3d), and $p_1(t)$ remained positive at least until the end of 2018. This anomalous wind forcing thus was likely the cause of the anomalous circulation that led to freshwater transport from the Icelandic shelf/slope region into the central Iceland Basin.

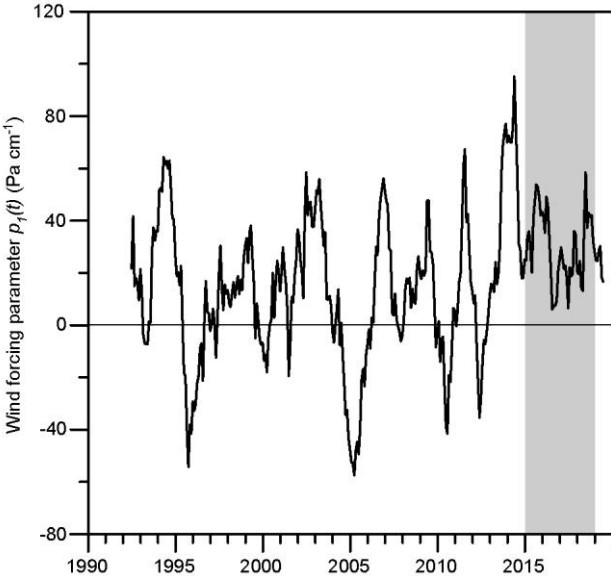

**Figure 11.** Twelve-month running mean of the wind forcing parameter $p_1(t)$ defined by Supplementary Eq. (S5). The shaded area indicates the 2015–2018 period.



*Data availability.* The time series of salinity of the upper ocean, shown in Fig. 1 were obtained from ICES (https://ocean.ices.dk/iroc/, downloaded July 2020). Data from OSNAP moorings M1, M2, M3, and M4 are available at
https://doi.org/10.7924/r42n52w51. The Extended Ellett Line data are available from https://www.bodc.ac.uk. Salinity data from Icelandic standard sections are available at the Marine and Freshwater Research Institute by request.

*Author contributions.* SRÓ, WJ, and NPH made data available. BH made the analyses and wrote the manuscript in cooperation with all the co-authors.

*Competing interests.* The authors declare that they have no conflict of interest.

*Financial support.* This study was supported by the European Union's Horizon 2020 research and innovation programme under grant agreement no. 727852 (Blue-Action). WJ was supported through NSF OSNAP grants OCE1259398 and OCE1756231. NPH and the Extended Ellett Line (now called the Ellett Array) were supported by the NERC National Capability programme CLASS (Climate Linked Atlantic Sector Science, NE/R015953/1).

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
