# Peer review of "An overlooked freshwater source contributed to the extreme freshening event in the eastern subpolar North Atlantic after 2014"

_Ocean Science, 2021_

## Author Comment (AC2)

Responses to RC1: 'Comment on os-2021-14', Anonymous Referee #1, 05 Mar 2021

**Overall response to both referees.** Our original response to the two reviews was that it would probably be a futile effort to submit a revised version of the manuscript as explained in our comment (**AC1**: 'Reply to both referees', Bogi Hansen, 27 Apr 2021). After reconsideration and correspondence with the editor, we have decided to resubmit and the following text details our responses to the original reviews and the revisions implemented. We acknowledge that the original text (and title) was probably open to misunderstanding. We have tried to make the revised version clearer as to the aims, analyses, and results of the study. We have not, however, included any additional types of data sources as suggested by the referees. As elaborated below, we do not believe that any of the suggested sources (satellite-salinity, ARGO-profiles, or ocean reanalyses) would provide substantially more convincing evidence than the sources that we discuss in the original manuscript.

**Responses to general comments from RC1**

**Referee comment: Lack of quantification**: From the outset, it was unclear what signal(s) the paper is attempting to explain. Holliday et al. (2020) describes a freshening of the upper 1000 m of the Iceland Basin and its forcing mechanisms, without mention of freshwater emanating from Iceland. In particular, Holliday et al. (2020) find a remarkable agreement between the magnitude of the salinity increase of the Scotian Shelf/Gulf of Maine region with the magnitude of the salinity decrease of the eastern SPG, implying that the whole of the freshening in the eastern SPG can be explained by freshwater coming from the Grand Banks region. Yet this paper seems to assume that the mechanisms presented in Holliday et al. (2020) are insufficient to explain the freshening. What evidence is there of this? How much of the freshwater signal from Holliday et al. (2020) is unexplained? And can the magnitude of the freshwater flux from Iceland explain the unexplained portion?

Response: It was never our intent to imply that the runoff from Iceland could account for most of the freshening in the eastern SPNA or even the Iceland Basin. Rather, we wanted to argue that the local and shallow origin of this freshwater source might have a disproportionate effect on the surface salinity of the Iceland Basin even if the runoff from Iceland is much smaller than the source described by Holliday et al. (2020). Clearly, this message has not been sufficiently well stated in the original manuscript. We have now changed the title (New title: "The potential role of Icelandic runoff in the extreme surface freshening event in the Iceland Basin around 2015") and the text, especially in the introduction (Sect. 1), but also in various places throughout the text to emphasize this. We have also re-organized the contents and modified the discussion, especially Sect. 4.1, which in the new version is split into two subsections (Sect. 4.1 and Sect. 4.2).

**Referee comment:** If indeed surface salinity is of interest (and it appears that it is), why not use satellite sea-surface salinity, which has near global coverage from 2009-present? From this, one could construct a time series of surface salinity for a given region, and then do an analysis of the relative roles of various mechanisms.

Response: To our knowledge, satellite sea-surface salinity data are not reliable close (50 km) to land (Sea Surface Salinity | NOAA CoastWatch & OceanWatch) and more generally are not very accurate (doi:10.1002/2014JC009961). We find extremely high scatter in these data for the Iceland Basin and have not succeeded in using them to trace the origin of the extreme surface freshening reported by Holliday et al. (2020) (their Fig. 4c).

**Referee comment: Results insufficient to support claims**

Response: The referee addresses each of our five main data sets and finds none of them supporting our hypothesis of a contribution from Icelandic runoff. Below, we respond to each of the points individually but, more generally, we agree that none of these data sets provides absolute proof of the hypothesis. We argue, however, that all of them may be interpreted to support it. Although we cannot prove the hypothesis, we find the collective evidence to support the likelihood of it. This ought to have been more clearly stated in the original manuscript. In the new version, it is reflected in the changed title and various places in the text.

**Referee comment:** If the water were to follow the SSH isolines indicated in either Fig 2 or 3a, then the fresh water would flow almost directly into the Irminger Sea and not affect the salinity of the Iceland Basin.

Response: It is certainly true that none of the isolines in Fig. 2 or Fig. 3a show a direct path from the south-Icelandic shelf into the central Iceland Basin, but neither of these figures nor any combination of them will reflect the flow path at any given time. Our argument is that during periods with a high principal component, the flow south of Iceland will have a stronger south-westward component. This should now be better emphasized by the added arrow in the new Fig. 4a and text.

**Referee comment:** there is no indication that this mode is at all related to the shelf-basin exchange (geostrophic or ageostrophic) around Iceland. ... Given these statements, the motivation for looking at altimetry to document shelf-basin exchange is unclear.

Response: As mentioned in the response to the comment above, we use the altimetry to address the open-ocean flow; not shelf-basin exchange. We acknowledge that the question of shelf-basin exchange was poorly addressed in the original manuscript. In the revised version, we have added a paragraph (and a new supplementary figure) to the text on drifters and to the discussion (Sect. 4.1).

**Referee comment:** Surface drifters – the authors use 11 surface drifter tracks that crossed from the Iceland shelf into the western Iceland Basin between 1995 and 2018 to demonstrate that Icelandic shelf water influences the western Iceland basin. But these drifters represent a small portion (1/8) of the total surface drifters that crossed into the Iceland shelf region during this period, thus the western Iceland basin is not a primary pathway for the freshwater around Iceland. There is the possibility that at certain times, this pathway is more important than others (potentially important between 2014-2018), but the authors acknowledge that "With only 11 drifters taking this path, it would be hard to determine temporal variations" (line 207). The authors then attempt to tie the surface drifter pathways to the altimetry EOF, but their argument falls flat because the altimetry EOF does not indicate shelf-basin exchange around Iceland (i.e. the motivation for using the drifters). Thus I believe the surface drifters demonstrate the opposite of what the authors contend: that the majority of the freshwater on the Iceland shelf does not flow into the Iceland Basin, and that it either goes eastward into the Nordic Seas or westward into the Irminger Sea.

Response: It was never our intent to claim that the western Iceland basin is a primary pathway for the freshwater around Iceland. On the contrary, we intended to argue that this pathway is not normal, but that it may have been more frequent during the freshening event when the principal component of the altimetry was high. Hopefully, this is clearer in the revised text.

**Referee comment:** The small number of drifters that flow into the western Iceland basin are a small percentage of a small freshwater flux from Iceland (~5 mSv, no reference salinity provided), and therefore likely represent a very small quantity.

Response: We agree that the contribution from Icelandic runoff to the western Iceland Basin under normal conditions is small. Even when the circulation favours this pathway, its contribution is likely small, but it may nevertheless affect the near-surface layer sufficiently much to reduce surface salinity substantially as we try to argue quantitatively in Sect. 4.1. The reason that no reference salinity is given for the 5 mSv value is that this is not calculated from seawater using a reference salinity, but is the amount of freshwater estimated to enter the ocean from Iceland as runoff..

**Referee comment:** Extended Ellett line – the authors contend that the salinity structure in the upper 200 m across the Iceland basin indicates an input from Icelandic-sourced freshwater. The salinity structure varies between years, and during periods of low salinity (2015-2017), the salinity of the central Iceland basin was lower than the eastern and western boundaries (in contrast to the other years, which increased almost monotonically from Iceland to Scotland). But again, this result seems to fly in the face of the authors' argument that the freshwater is coming off Iceland – if the input from the Iceland shelf is confined to the western Iceland basin, as indicated by the surface drifters, then shouldn't the salinity of the western Iceland basin decrease the most? The low salinities in the central Iceland basin (in regions of northward velocities) indicates that these low salinity waters are sourced from the south, rather than the north.

Response: We did not intend to claim that "the input from the Icelandic shelf is confined to the western Iceland basin" and most of the drifters that survive sufficiently long return to a more "normal" flow pattern. This emphasizes that the flow pattern that brings Icelandic runoff into the western Iceland Basin varies in strength and dominance (Fig. 3d). We have enlarged the original Fig. 4 so that this is clearer. This variation also has a seasonal component (Fig. 3c) which is strongest in winter, whereas the EEL was occupied in summer. We have added a new panel to Fig. 5 and some explanatory text that hopefully helps understanding our argument, but we recognize that our interpretation is rather speculative.

**Referee comment:** OSNAP moorings – the argument here is that the high-frequency variability in the vertical salinity structure cannot be explained by far-field forcing from the western boundary and instead requires more local forcing (line 300). However, I do not follow this logic. The Iceland basin is full of eddies and small-scale structure that advect property anomalies – why can't these cause the high-frequency salinity variability seen at the OSNAP moorings? If the argument from Holliday et al. (2020) were that a 'pool' of freshwater came off the Newfoundland/Labrador Shelf and moved coherently into the eastern SPG, then I would agree with the authors that the vertical structure of this salinity anomaly

would be eroded by the time it arrived to the eastern SPG. But I don't believe that's the argument in Holliday et al. (2020).

Response: We agree that the OSNAP data exhibit considerable high-frequency salinity variability, but our argument was aimed at variations on longer time-scales. Thus, the enhanced salinity gradients in the uppermost 200 m in Fig. 6 have time-scales exceeding those that are normally associated with the meso-scale, especially for site M2. We have modified the text to address this.

**Referee comment:** Hydrographic sections south of Iceland – I found the timeseries of freshwater thickness compelling, but given the infrequent coverage of the hydrographic sections, these data should only be considered ancillary, and not central to the argument. In other words, I would be more convinced of the authors' arguments presented in this section if the altimetry and drifter data were stronger. Furthermore, the lack of velocity data diminishes what can be inferred from these sections, particularly regarding the strength of the freshwater fluxes. It was unclear to me even which direction these freshwater fluxes at each line are directed considering that Section IH "seems to be located in a divergence zone between eastward- and westward-flowing waters" (line 276). Does this imply that the freshwater flux at Section ST is eastward? How does that align with Fig. 1a? Response: The statement cited "Section IH seems to be located in a divergence zone between eastward- and westward-flowing waters" was based on the drifter tracks, but they mainly represent the flow off the shelf and the evidence for the coastal current does not support this (cyan-coloured arrows on the new Fig. 7 and references). Thus, this statement was probably misleading and has been removed. Thank you for noting this.

**Referee comment:** And what evidence is there to multiply the salinity fields by a constant 10 cm/s velocity (line 273)? Response: The text has been re-written and a new reference added.

**Specific Referee comment 1:** Line 48-49 "most of the entrainment into overflow also occurs in the eastern SPNA" are you referring to most of the water mass transformation across the isopycnals of maximum overturning?

Response: A new reference has been added.

**Specific Referee comment 2:** Line 63 where is the Faxafloi line? Please mark this on Fig 1a. Response: The Faxaflói 9 standard station is marked on Fig. 1a. Additional text has been added to clarify this.

**Specific Referee comment 3:** Line 85 – what is the reference salinity for the 5 mSv of freshwater carried by the Iceland Coastal Current?

Response: We did not claim that the Coastal Current carries 5 mSv of freshwater, but rather that it receives 5 mSv of Icelandic runoff. Since this is pure freshwater, a reference salinity is not needed or appropriate.

Specific Referee comment 4: Line 87 – "In addition to this" – what does "this" refer to?

Response: "this" has been changed to "the runoff".

**Specific Referee comment 5:** Section 2 – Is it necessary to have so many subpoints? Can all the data be summarized in a single paragraph? Response: All the oceanic data have now been summarized in one subsection.

**Specific Referee comment 6:** Line 131 – it would be instructive if Supplementary Table S1 also included the seasonal and interannual timing of these cruises.

Response: Table S1 is a list of standard stations, not cruises. Information on cruises is in the original Sect. 2.3.3 (new Sect. 2.1) and we find that the table would become very complex if more detail on cruises were to be added to it.

**Specific Referee comment 7:** Line 148 – Please elaborate on "This makes the result independent of any assumptions used in generating the MDT...". To what assumptions are you referring? Response: This sentence has been deleted.

**Specific Referee comment 8:** 2 – If the black line delineates the contour that separates the flow north and south of the Faroes, why doesn't it intersect the Faroes? Response: The thick black line has been removed (new Fig. 3).

**Specific Referee comment 9:** Line 154 and elsewhere – replace "anticlockwise" with "cyclonic" (or "clockwise" with "anticyclonic"). Response: Has been done.

**Specific Referee comment 10:** Line 157 – what is meant by 'distorted'? The average flow pattern incorporates all time scales, including synoptic variability. The horizontal resolution of the SSH will not resolve the mesoscale, but why does that mean it's distorted? Response: This sentence has been deleted..

**Specific Referee comment 11:** Line 165 – what is meant by long time scales? Fig 3c outlines the seasonal component of this mode... does the seasonality come into the argument at all? Response: "long time scales" has been replaced by "seasonal as well as inter-annual time scales"

**Specific Referee comment 12:** 3c – it would be useful to use a box-and-whisker plot here to show the median, quartile ranges, outliers, etc. Currently, it is tough to determine the strength of the seasonal cycle purely from the overlapping markers.

Response: We have added a curve showing monthly average and a shaded area representing average ± standard error.

**Specific Referee comment 13:** 3d – is this panel a zoom in on panel b? If so, this should be highlighted in panel b (maybe draw a rectangle in panel b around the bounds of panel d, or alternatively just make a note of this in the caption for panel d).

Response: A shaded rectangle has been added to panel b and text added to the caption to emphasize that panel d shows 28-day averaged (not low-passed) principal component (as does panel b).

**Specific Referee comment 14:** I Line 175 – It should be noted that this sentence refers to reduced cyclonic circulation, rather than an actual anticyclonic circulation in the Iceland basin. Response: Has been done.

**Specific Referee comment 15: Line 210 – why is only longitude considered?**

Response: We have redone the analysis more carefully, considering both latitude and longitude and with a more precise area definition (new supplementary Fig. S7). This increased the absolute value of the correlation coefficient from 0.37 to 0.45.

**Specific Referee comment 16:** Lines 207-214 – the argument that the altimetry and surface drifters are well correlated would be significantly strengthened if there were an accompanying figure. Can you produce a set of maps of altimetry and surface drifter tracks for ~4 time steps? Or alternatively show the two curves that are correlated? I am surprised that there is such good agreement between these data sets considering that the Ekman velocities are not added into the surface geostrophic velocity field. Would these comparisons improve if the Ekman velocities were added? Response: We have not been able to produce such a map in any way that makes it give useful information. And we have not tried to calculate Ekman velocities since we do not believe that this would help for the present study, although we agree that such an analysis would be interesting more generally.

**Specific Referee comment 17:** Line 230 – remove comma after 'months'. Response: Done.

**Specific Referee comment 18:** Figure S7b – at mooring M2, why does the salinity decrease from 50 m to 100 m when it increases in panel a and Fig. 6b? Response: Due to data loss, the salinities at 50m and 100m depth in Fig. S7b are averaged over different periods as noted in the figure and the caption.

**Specific Referee comment 19:** Lines 250-253 – It is not clear why the freshwater content is valid here. Schauer and Losch (2019) discuss the arbitrary use of reference salinities in the calculation of freshwater transport, and their arguments hold whether the freshwater came from 'pure freshwater' or not. The only way around this issue is to use a closed volume in which the mass budget is balanced. Another method is to report the freshwater fluxes relative to two reference salinities to demonstrate their sensitivity to the choice of reference.

Response: Our use of the term "reference salinity" was unfortunate since our method for deriving freshwater transport is different from the method criticized by Schauer and Losch (2019). The term has now been deleted and the text hopefully clarified.

**Specific Referee comment 20:** 7 – why is 200 m chosen as a bottom limit? Response: A note has been added that this is the typical depth of the shelf edge.

**Specific Referee comment 21:** Line 285 – There is no evidence presented here that the drifters followed different paths in different years, and the authors admit as much (line 207). Response: This sentence has been deleted.

**Specific Referee comment 22:** Line 351 – how is the importance of the precipitation trend assessed relative to other mechanisms? Given that there is no explanation of what signal the authors are trying to explain, it is hard to assess whether the precipitation is small or large comparatively. Response: This sentence has been deleted..

**Specific Referee comment 23:** Line 356 – This explanation of the errors in the MDT near land and steep topography should go in section 3.1. This is much more clear than the current explanation on lines 147-149.

Response: Has been done.

**Specific Referee comment 24:** Line 361 – Two drifters followed this pathway in 2015 from table S2, but earlier the authors admit that they cannot use the drifters for temporal variability due to insufficient coverage. Along these lines, there were three drifters that followed this pathway in 1996 when the principal component was near zero.

Response: We do not claim a one-to-one correspondence between drifters and altimetry, but the highly significant correlation does argue for some correspondence.

**Specific Referee comment 25:** Line 364 – "we find it most likely that the Icelandic freshwater source was the primary cause of the near-surface freshening events in the 2015-2017 period." I don't know of any evidence to support this claim, particularly in regard to the ranking of roles that various mechanisms played.

Response: This sentence was badly phrased and might be easily misunderstood. It has been deleted.

---

## Author Comment (AC3)

**Responses to RC2**: 'Comment on os-2021-14', Anonymous Referee #2, 13 Apr 2021

**Overall response to both referees.** Our original response to the two reviews was that it would probably be a futile effort to submit a revised version of the manuscript as explained in our comment (**AC1**: 'Reply to both referees', Bogi Hansen, 27 Apr 2021). After reconsideration and correspondence with the editor, we have decided to resubmit and the following text details our responses to the original reviews and the revisions implemented. We acknowledge that the original text (and title) was probably open to misunderstanding. We have tried to make the revised version clearer as to the aims, analyses, and results of the study. We have not, however, included any additional types of data sources as suggested by the referees. As elaborated below, we do not believe that any of the suggested sources (satellite-salinity, ARGO-profiles, or ocean reanalyses) would provide substantially more convincing evidence than the sources that we discuss in the original manuscript.

**General referee comments from RC2**
**Referee comment:** This paper uses observational data (satellite altimetry, trajectory of surface drifters, in situ salinity measurements) to suggest an overlooked freshwater source off the Iceland coast. The authors argue that this addition of low-salinity waters likely contributed to the recent freshening in the eastern subpolar North Atlantic that was described by a previous study (Holliday et al. 2020). Despite the title referring to "freshwater source", the manuscript does not argue for a source of freshwater, but rather a phenomenon where fresher surface waters off the coast of Iceland were diverted into the Iceland Basin due to an anomalous circulation pattern south of Iceland.
Response: The title has been changed (New title: "The potential role of Icelandic runoff in the extreme surface freshening event in the Iceland Basin around 2015") to emphasize that we are referring to runoff from Iceland.
The study includes a comprehensive analysis of various data sets and observations and I acknowledge the attempt to connect these diverse datasets with each other. However, these results do not convincingly show the relevance of such a freshwater source to contribute to the overall freshening. Therefore this complicates without suitable justification the story that Holliday et al. (2020) laid out to explain the freshening.
Response: Our intent was not to explain the "overall freshening", but rather the enhanced freshening of the surface layer. Hopefully, this is clearer in the revised text.

**Referee comment:** The main issue with this study is the lack of quantification. This type of analysis essentially requires a budget analysis in order to provide a clear freshwater estimate and establish that the redirection of low-salinity water off the Iceland shelf is a relevant signal in the eastern subpolar North Atlantic.
Response: With Table 1 and the discussion in Sect. 4.2, we did quantify the contribution from the Icelandic runoff to the freshening of the Atlantic inflow to the Nordic Seas and it was indeed small (original lines 409-412). As stated above, our main aim was, however, to explain the enhanced freshening of the surface layer relative to deeper waters. With the discussion in the original Sect. 4.1, we believe to have done that quantitatively. In the revised version, we have tried to emphasize this in Sect. 4.1 and Sect. 4.2.

**Referee comment:** The analysis presented here does not meet current standards given that there is now readily available data that goes beyond mooring and hydrographic ship data. In particular it is surprising that Argo profiling data has not been utilized to complement the other salinity observations. Especially for recent time periods, salinity profiles from Argo floats should provide a better picture of the spatiotemporal variability in that regions. There are also a number of gridded salinity products derived from Argo and other profile data as well as satellite-derived surface salinity estimates which are publicly and readily available. In particular, the use of ocean reanalysis products (e.g., SODA, ECCOv4) would be essential in such investigation as these allow closed budget analyses that can establish underlying mechanisms. It is essential to include these analyses, which then can be compared to the present hydrographic data to see if the picture is still consistent with the freshening and hypothesized pathway of freshwater over the Iceland Basin.

Response: Using data from profiling ARGO floats might be worthwhile, but we do not find it likely that the relatively few ARGO floats that at any time are present in the Iceland Basin with typical parking depths of 1000 m and high spatio-temporal variability can provide much more convincing evidence. As to ocean reanalysis products, their quality will depend not only on the observational data, but also on the model used. For the present purpose, we would need a model with sufficient resolution and realism to describe the flow over the Icelandic shelf and the shelf-basin exchange in a realistic manner. We do not know of any existing reanalysis product that satisfies this criterion.

**Specific referee comments**
**Title:** The title is misleading since it describes a "freshwater source", but in fact the hypothesized mechanism is a change in circulation.

Response: The title has been changed to: "The potential role of Icelandic runoff in the extreme surface freshening event in the Iceland Basin around 2015".

**Line 34**: What is meant by the term "buffering region"?

Response: The term "buffering" has been changed to "transition"

**Lines 43-46**: This statement needs references. I am not aware that the subpolar North Atlantic is nutrient limited and would expect it to be largely light limited. Thus, a reduction in vertical mixing could also lead to higher productivity due to a decrease in light limitation.

Response: Elsewhere in the original manuscript, we quote references suggesting that the central Iceland Basin (not the SPNA as a whole) may be oligotrophic. Here, the text was intended as a general statement to justify the importance of stratification for biological production. And we agree that stratification may increase productivity, which was the intent with the first part of that sentence: "increased stratification may affect instantaneous primary production".

**Section 2**: I think this section would read better without the different subsection and instead have a single section with complete paragraphs. Also, I feel there is a lot of important information missing. For example, the EOF analysis is a key method in this study, but it is only described in the supplementary material. A shortened description should be included in the Method section too. As well, what is the

calculation method for freshwater thickness? What analysis software has been used (Python, Matlab etc.)?

Response: Subsections in Sect.2 have now been combined. The EOF analysis is a standard method and we did not find it necessary to describe it in detail. In the revised version, we have added somewhat more detail and changed the reference from Preisendorfer (1988) to a more well-known methodology text (Emery and Thomson, 1998).

**Line 128-129**: How has potential density been derived?

Response: In the standard way by using potential temperature and referring to the surface (Emery and Thomson, 1998).

**Line 130-134**: The location of these sections should be included in the map (Figure 1).

Response: Fig. 1a is already rather crowded and the sections are shown on Fig. 7. A reference to that figure has now been added to the caption of Table S1.

**Line 135-137**: This is confusing phrasing. ERA5 should be stated as the atmospheric reanalysis product and CDS the repository from which the data has been obtained.

Response: We used the text recommended by the data supplier

**Section 3**: Instead of calling the subtitles by the data product, it would be better to have an actual subject that refers to  the finding/processes etc. For example, instead of the title "satellite altimetry", it should be called something like "geostrophic flow pattern".

Response: We found it difficult to find alternative headings for all the subsections and have not done this.

**Line 157**: Need to explain more how the spatial pattern in Figure 2 can be interpreted to identify the flow.

Response: The beginning text of this section (new Sect. 3.2) has been modified to: "In the geostrophic approximation, the slope of the sea surface is proportional to the speed of the surface layer".

**Line 159-163:** I think the description of the EOF analysis fits better in Section 2.

Response: We have added some more text on the EOF method to Sect. 2, but have kept the results from the analysis in Sect. 3.

**Line 160:** Please specify the spatial domain that has been used in the EOF analysis (cite lat and lon bounds).

Response: Done

**Line 166**: "Anomalous behaviour" is unnecessarily vague. Just describe what is anomalous about it.

Response: Has been done.

**Figure 3 b-d**: Units are missing on the axis labels. Is it "cm" as in panel a?

Response: The product of the spatial mode and the principal component should have the same dimension as the original data set, that is cm (e.g., Eq. S1). As explained in the Supplementary methods, we assign the dimension to the spatial mode, which means that the principal component is dimensionless.

**Lines 174-176**: How is the connection made between zero to negative PC values and enhanced clockwise circulation? This connection has not been clearly explained.
Response: In the new version, an arrow is added to the new Fig. 4a and explanatory text added to the caption.

**Line 203**: I'm not sure if the term "streamlines" is appropriate here. This would set the EOF equivalent to a stream function, which I don't think is the case.
Response: This term is no longer in the manuscript.

**Lines 211-213**: It is not obvious to me how a fairly weak correlation between the zonal drift and PC time series leads to that conclusion. Maybe it helps if the authors actually describe the process behind the apparent correlation.
Response: The correlation has been improved in the new version and the text clarified.

**Lines 224-226**: The fact that the drop in salinity is confined to the middle of the basin indicates to me that the freshening is sourced from the south, as this region corresponds to the pathway of the northward flowing NAC.
Response: The text has been modified.

**Lines 248-249, Fig 7c**: How's the freshwater content calculated? This needs to be included in Section 2.
Response: The explanation is hopefully clearer now

**Line 254**: Cite the actual value that is chosen as the reference salinity.
Response: As explained in the revised version, the term "reference salinity" was misleading in this context. The new text should be clearer.

**Line 274**: The freshwater flux of 5 mSv needs to be put in context. This is quite small compared to any other freshwater flux estimate over the subpolar North Atlantic.
Response: In the revised version, this point is emphasized in the Introduction, as well as in Sect. 4 and Sect. 5.

**Lines 276-277**: Please clarify the relevance of this statement.
Response: The text has been modified.

**Section 4**: A majority of the content of this section can be regarded as results. Usually, the Discussion is for bringing up previous studies, raising potential concerns and caveats, and restating the main findings.
Response: The first part of Sect. 4 has been completely re-written and hopefully clarified.

**Lines 296-299**: If the freshening is explained by just deviation in the circulation, then shouldn't we expect salinification in the downstream region where the Icelandic source usually ends up?
Response: That may well be, and we have added a sentence to that effect in the new Sect. 4.3, but we do not have the data to assess that more quantitatively.

**Line 305**: This statement need to be supported by quantitative analysis.
Response: This statement (and the accompanying old Fig. 8) have been removed.

**Line 308**: The phrasing "more normal flow" is odd. It should be clarified what normal is.
Response: This sentence has been removed.

**Line 324**: The phrasing "drained of freshwater" does not make sense in the context of oceanic freshwater content.
Response: The text has been modified and the term "drainage" removed.

**Line 332**: Clearly describe the steps used to get to the estimate of 0.5 m.
Response: This is hopefully clarified in the revised text (new Sect. 4.2).

**Section 4.3** This suggest only a minor influence of the Icelandic freshwater source and thus contradicts the whole premise of this study.
Response: We assume that the referee means the old Sect. 4.2 (not 4.3) and we never intended to claim that Icelandic runoff was the main cause of the overall freshening of the eastern SPNA or the Iceland Basin. Only that this could explain the surface freshening became so extreme. This section has been substantially re-written (and abbreviated).

**Line 415**: Statement "improves conditions for primary production" needs references.
Response: The text has been rephrased.

**Line 427**: Specify a quantitative criterion to determine convection depth.
Response: The text has been rephrased and a precise definition is provided for the values in Table 2.

**Page 20**: This is a strange way to end the paper. A section containing Conclusions is missing.
Response: A new Conclusions section (Sect. 5) has been added.